# Microreserves are an important tool for amphibian conservation
Emma Steigerwald [1,2] ✉, Julia Chen[1,2,6], Julianne Oshiro[1,2,6], Vance T. Vredenburg[1,3] ✉,
Alessandro Catenazzi [4,5] & Michelle S. Koo [1] ✉

Initiatives to protect 30% of Earth by 2030 prompt evaluation of how to efficiently target shortcomings in the global protected area (PA) network. Focusing on amphibians, the most vulnerable vertebrate class, we illustrate the conservation value of microreserves, a term we employ here to refer to reserves of <10 km$^2$. We report that the network continues to under-represent threatened amphibians and that, despite this clear shortcoming in land-based conservation, the creation of PAs protecting amphibians slowed after 2010. By proving something previously assumed–that amphibians generally have smaller ranges than other terrestrial vertebrates–we demonstrate that microreserves could protect a substantial portion of many amphibian ranges, particularly threatened species. We find existing microreserves are capable of hosting an amphibian species richness similar to PAs 1000–10,00X larger, and we show that amphibians' high beta diversity means that microreserves added to a growing PA network cover amphibian species 1.5—6x faster than larger size categories. We propose that stemming global biodiversity loss requires that we seriously consider the conservation potential of microreserves, using them to capture small-range endemics that may otherwise be omitted from the PA network entirely.

As the world unites behind efforts to mitigate the effects of the sixth mass extinction[1] by protecting 30% of the earth's surface by the year 2030, a goal known as "30 × 30"[2], we are at a pivotal moment to evaluate land-based conservation planning. Key questions include where to expand the protected area (PA) network, as well as how to balance the size versus the number of new PAs. More than 15% of the earth's terrestrial surface is already protected[3], but the existing PA network is inadequate in representing biodiversity– particularly threatened biodiversity[4–6]. These deficiencies have multiple causes. First of all, the earliest PAs were created to protect scenic landscapes and wildlife, safeguard natural resources, and provide recreational opportunities, rather than to sustain biodiversity, with various motivations for PA creation persisting to this day[7]. Next, although the PA network has grown to include biodiversity-motivated PAs, there are always economic, social, and political constraints affecting PA placement[8]. Finally, there is strong taxonomic bias in how well species are represented by the modern PA network. The disparity between taxa is partly due to more conservation attention being focused on charismatic megafauna, and to their use as surrogates for all biodiversity in PA design[9,10]. However, some taxa also just have smaller range sizes, higher endemicity, and distinct distributional patterns[11], resulting in a higher likelihood that they will be passively excluded from PAs[12]. The existing PA network serves amphibians particularly poorly, such that they are the most underrepresented class of terrestrial vertebrates[6,13–15]. In some regions, the PA network does not represent amphibian diversity better than if PAs had been placed by chance[5]. In other regions, the existing PA network is placed entirely contrary to patterns of amphibian endemism[16].

Like insects or freshwater mollusks[17,18], amphibians are undergoing global declines and extinctions, with habitat loss serving as a major driver[19], yet are highly unlikely to be the focal taxa of new PAs[20–23]. Though amphibians have existed on earth for nearly 400 million years[24], in just the last decades there have been an alarming number of extinctions. 37 species are confirmed to have gone extinct, with as many as 185 additional species possibly extinct, within the last 150 years[25]. Meanwhile, more than 43% of species have populations that are declining[26]. Current declines set amphibians on track for extinction rates exceeding those estimated for previous mass extinctions[27]. Since the pace of habitat conversion is accelerating[28], PA designation will be critically important to attenuate a new planetary mass extinction event. If we are to meet 30×30 goals, we must expand the current

[1]Museum of Vertebrate Zoology, University of California, Berkeley, Berkeley, CA 94720, USA. [2]Department of Environmental Science, Policy, and Management, University of California, Berkeley, Berkeley, CA 94720, USA. [3]Department of Biology, San Francisco State University, San Francisco, CA 94132, USA. [4]Department of Biological Sciences, Florida International University, Miami, FL 33199, USA. [5]Centro de Ornitología y Biodiversidad, Lima, Peru. [6]These authors contributed equally: Julia Chen, Julianne Oshiro. ✉e-mail: emma.c.steigerwald@gmail.com; vancev@sfsu.edu; mkoo@berkeley.edu

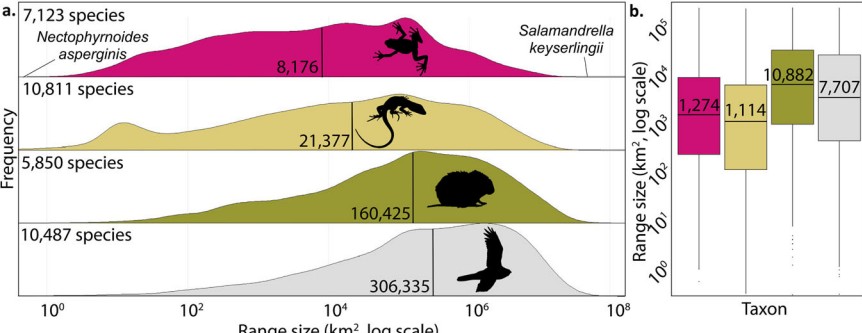

**Fig. 1 | Comparison of range sizes for terrestrial vertebrate classes. a** Smoothed density histogram for terrestrial vertebrate range sizes: amphibians, birds, reptiles, and mammals. Only the terrestrial range area of each species is considered. The median range size of each taxon is marked with a vertical black line. For amphibians, the Kihansi spray toad (*Nectophrynoides asperginis*, 0.104 km²) has the smallest range and the Siberian newt (*Salamandrella keyserlingii*, 14,700,000 km²) has the largest range. The number of species for which data was available in each taxonomic group is indicated in each panel. **b** Range size (in km², log scale) of threatened and extinct species in each taxon. In this boxplot, the horizontal line represents the median, the box represents the interquartile range, and the whiskers represent the range of values within 1.5 times the interquartile range from the first and third quartiles. The number of threatened and extinct species for which data was available in each taxonomic group was $n_{birds}$ = 3935; $n_{amphibians}$ = 4585; $n_{reptiles}$ = 3819; $n_{mammals}$ = 3752.

PA network by an additional 22 million km² in the next 7 years, providing an incredible opportunity to think explicitly about how we might shore up the shortcomings of the current PA network for those taxa it is currently failing[18,19,29].

Classical studies in ecological theory predict that biodiversity value increases with PA size[30,31], such that the conservation potential of establishing small protected areas is likely to be discounted. We propose that failing to consider small protected areas as critical conservation tools ensures that our global PA network will entirely exclude many small-range endemics, as typified by much of amphibian diversity. In fact, the conservation of many amphibian species can be effective at small spatial scales[32,33]--with much of amphibian diversity having high beta diversity, limited dispersal, and philopatric behaviors[34–36]. Here, we explore the idea that strategically-placed reserves of 10 km² or less, here termed as 'microreserves', could drastically enhance the value of the PA network for amphibians. First, we confirm that amphibians generally have smaller ranges than other terrestrial vertebrate classes, as is often assumed. Then, we provide an up-to-date assessment of amphibian coverage provided by the global PA network, using expert-curated range maps for more than 83% of amphibian diversity (7094 of 8498 recognized species)[37], including 778 new amphibian species and 121,505 new PAs relative to the last time a similar assessment was undertaken[12]. We show that amphibians continue to be underrepresented by the global PA network. Despite signs that PAs are being placed more strategically over time in places of higher amphibian vulnerability, we find that fewer amphibian-containing PAs are being created over time, such that the rate at which amphibian diversity is being integrated into the global PA network has recently lagged. Promisingly, we demonstrate that it is possible for microreserves to host amphibian species richnesses rivaling those of the world's largest PAs, and show that the PA network's coverage of amphibian diversity can be more rapidly augmented through the addition of microreserves than larger PAs. Together, we use our findings to argue that effective amphibian conservation will require that we not discount the conservation potential of new microreserves, which should be deployed strategically to capture small-range species that will otherwise get left behind in land-based conservation efforts.

## Results
### Amphibians have smaller ranges than other terrestrial vertebrates
We assembled species-specific geographic range area maps from 31,828 species, including all classes of terrestrial vertebrates: 7094 amphibians from AmphibiaWeb and the International Union for the Conservation of Nature, or IUCN[26,37]; 10,811 non-avian reptiles from the IUCN and the Global Assessment of Reptile Distributions group[26,38]; 5850 mammals

from the IUCN[26]; and 10,487 birds from BirdLife International[39]. Amphibians had a smaller median range size than other vertebrates classes in all pairwise tests (Fig. 1a; Supplementary Table 1; Wilcoxon rank sum test, $p < 0.001$). When we compared range sizes of only threatened species (as determined by the IUCN Red List) between taxa, threatened amphibians also had significantly smaller median ranges than threatened birds and mammals in pairwise analyses (pairwise Wilcoxon rank sum test; p < 0.001; $n_{birds}$ = 3935; $n_{amphibians}$ = 4585, $n_{reptiles}$ = 3819, $n_{mammals}$ = 3752). In fact, one microreserve (<10 km²) would be sufficient to protect the majority of the distributional range of each of 140 endangered amphibian species. Finally, within class Amphibia, threatened species had a smaller median range size than non-threatened species (Wilcoxon rank sum test; $p < 0.001$).

### The rate at which amphibian-containing PAs are created is declining, but much amphibian diversity is still excluded from the PA network
The rate of new PA establishment for all amphibian-containing PAs and for amphibian-containing microreserves increased almost monotonically until the early 2000s (Fig. 2a), corresponding with a steady increase in the cumulative number of amphibian species covered by the global PA network (Fig. 2a). However, since the early 2000s the rate of amphibian-containing PA establishment has dropped, echoed by a decline in the rate of new amphibian-containing microreserves established. Despite this recent decline in new amphibian-containing PAs, the cumulative number of amphibian species protected by the network continues to increase– though the rate of gains has slowed since 2015 (Fig. 2a).

Over time, we also see that more PAs are being established in zones of high amphibian vulnerability (Fig. 2b). The best-supported model of how the proportion of threatened amphibian species in a PA responds to PA characteristics (latitude, ln(area), year of establishment, and IUCN protected area management category) retained latitude and the interaction between latitude and logarithmic area (Supplementary Table 2), with all coefficients being significant ($p < 0.001$). The most important characteristic is latitude, where the coefficient corresponds to a 6.7% ($e^{-0.07}-1$) decrease in the proportion of threatened species per degree moved away from the equator, while the interactive term is associated with a relatively minimal impact.

Although 97.3% (241,000 of 247,785) of PAs with a terrestrial component overlap with at least one amphibian range, almost 15.7% of amphibian species (1115 species) are left unprotected by the existing network (Fig. 3; henceforth referred as "unprotected species"). There is a higher proportion of threatened and extinct species (T&E species) among amphibians unprotected by the current PA network (35.8%; 400 species) compared to species protected by the current network (29.6%; 1771 species).

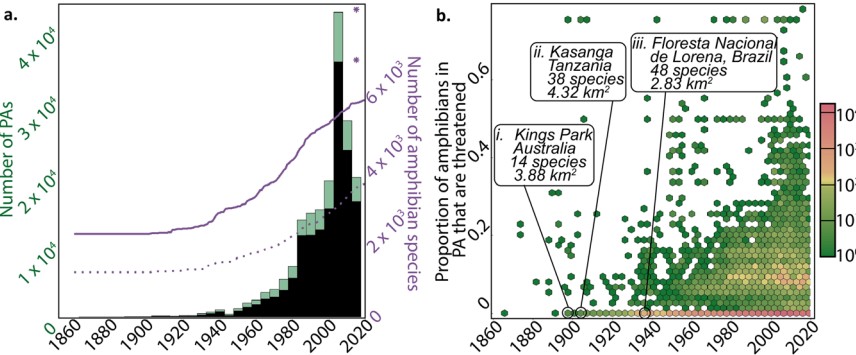

**Fig. 2 | Trends in PA placement over time. a** Counts of amphibian-containing PAs established over time (lefthand y-axis) and how that relates to the cumulative number of protected amphibian species (righthand y-axis). The green bar plot shows counts of PAs established over time, binned by 5-year units, and the black bar plot shows the same thing for microreserves only (area <10 km²). The solid purple line shows the cumulative count of protected amphibians over time as the PA network grew, and the dotted purple line shows the same thing for microreserves only (area <10 km²). The lower star represents the total number of amphibians in this study with spatial data (*n* = 7094), and the higher star represents total amphibian species described (*n* = 8489). **b** A hexbin heatmap of microreserves, showing the proportion of amphibian species in a microreserve that are threatened today (y-axis) relative to the year that microreserve was established (x-axis). Counts of microreserves are displayed with the color scale (z-axis). Old microreserves from three continents with a low (zero or near-zero) proportion of IUCN-threatened species are annotated (i, ii, & iii). Old microreserves from three continents with high species richness but no threatened species are identified.

**Fig. 3 | Conservation status of amphibians protected and not protected by the current PA network.** The proportion of species assigned each IUCN conservation status among amphibians either **a** overlapping in range or **b** not overlapping at all with the global protected area network.

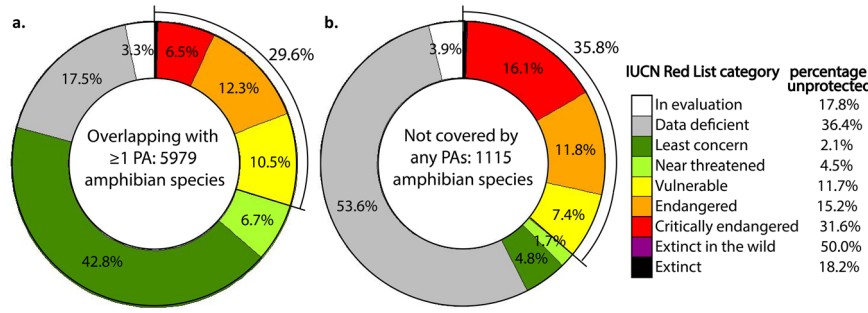

| IUCN Red List category | percentage unprotected |
|---|---|
| In evaluation | 17.8% |
| Data deficient | 36.4% |
| Least concern | 2.1% |
| Near threatened | 4.5% |
| Vulnerable | 11.7% |
| Endangered | 15.2% |
| Critically endangered | 31.6% |
| Extinct in the wild | 50.0% |
| Extinct | 18.2% |

Likewise, data deficient species (DD species) and species that have not yet been assessed are overrepresented among unprotected species (57.5%; *n* = 641 species), compared to only 20.8% among protected species (1244 species).

## PA networks cover amphibian diversity more rapidly through the addition of smaller PAs, which can rival the species richness of PAs orders of magnitude larger

The best-supported model of how total amphibian species richness in a PA responded to PA characteristics included all significant (*p* < 0.001) terms (Table S3): logarithmic PA area, PA establishment year, and their interaction; latitude; and IUCN protected area management category. Latitude is associated with a 3.3% ($e^{-0.033} - 1$) decrease in amphibian richness for each degree of latitude moved away from the equator, while logarithmic area, establishment year, and their interaction have a relatively negligible impact. Although most microreserves (<10 km²) are currently located in areas of low amphibian richness (Fig. 4a), we find that microreserves are also able to capture areas of high richness (annotations i, ii, & iii). Encouragingly, amphibian-rich microreserves that were established 90 or more years ago may still maintain a 0% proportion of threatened amphibian species (Fig. 2b: annotations i, ii, & iii). If we grow a PA network by iteratively sampling from the existing database, we find that the cumulative amphibian diversity covered by a network increases about 6x faster when microreserves are sampled than when PAs 10,000–100,000 km² larger are sampled, and 1.5x faster than when PAs 10–100 km² larger are sampled (Fig. 4b). Thus, amphibian species diversity included in a PA network is maximized through the addition of many microreserves rather than through an equivalent

geographic area contributed by only a few large PAs (Fig. 4b; largest reserves 10,000–100,000 km²).

## The distribution of PA sizes and coverage of amphibian diversity provided by the PA network varies regionally

Across different geographic regions, the size distribution of PAs, total amphibian species richness, and the proportional representation of threatened species among protected and unprotected amphibians varied greatly. Europe had the smallest median PA size (Fig. 5, 0.27 km²) but also no amphibian species whose range does not overlap with its PA network. Madagascar had the largest median PA size (270.40 km²) and only four species that do not overlap PAs (1.2%). The regions with the highest proportion of unprotected species were the islands of Melanesia, Micronesia, and Polynesia (41.4%; 123 species), while South America and Asia had the highest number of unprotected species (349 and 246 species, respectively). Central America, Mexico, and the Caribbean had the highest number of threatened and unprotected amphibian species (107 species), as well as the largest differential between the proportion of threatened species that are protected versus unprotected (27.5% more threatened species among unprotected than protected species).

PA establishment occurs at the scale of the country, and the most amphibian-rich country is Brazil (944), followed by Colombia (810), Peru (566), and Ecuador (520; Supplementary Fig. 1a). The country with the highest number of threatened amphibians is Colombia (233), followed by Mexico (227), Ecuador (184), and Madagascar (134; Supplementary Fig. 1b). The country with the highest number of unprotected amphibian species is China (156), followed by Papua New Guinea (122), India (111), and Mexico (99; Supplementary Fig. 1c). Colombia, Peru, Ecuador, China,

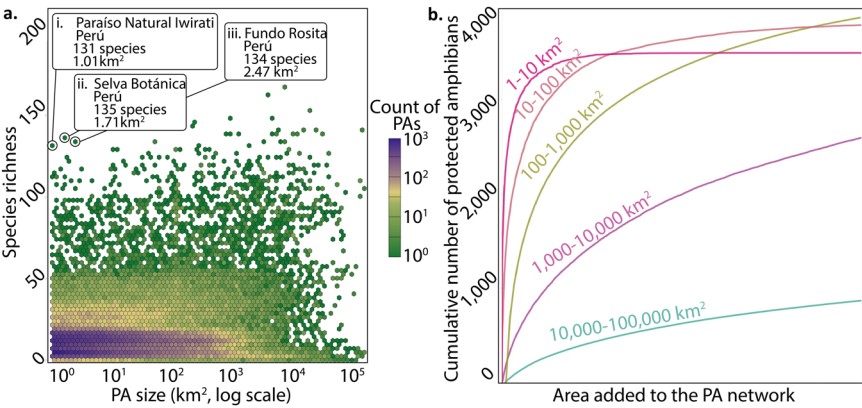

**Fig. 4 | Relationship between amphibian species richness and PA size, and amphibian species accumulation across five different PA size categories. a** How PA size relates to the amphibian species richness it contains. Microreserves with the highest species richness are identified (i, ii, & iii). **b** Cumulative proportion of protected amphibian species as you sample PAs of each size category (0–10 km², $n$ = 208,496; 10–100 km², $n$ = 27,762; 100–1000 km², $n$ = 11,975; 1000–10,000 km², $n$ = 3045; and 10,000–100,000 km², $n$ = 430) drawn from the WDPA database. The x-axis is scaled such that it represents equivalent area protected, regardless of the PA size category considered. The cumulative number of amphibian species with range data available was 7094.

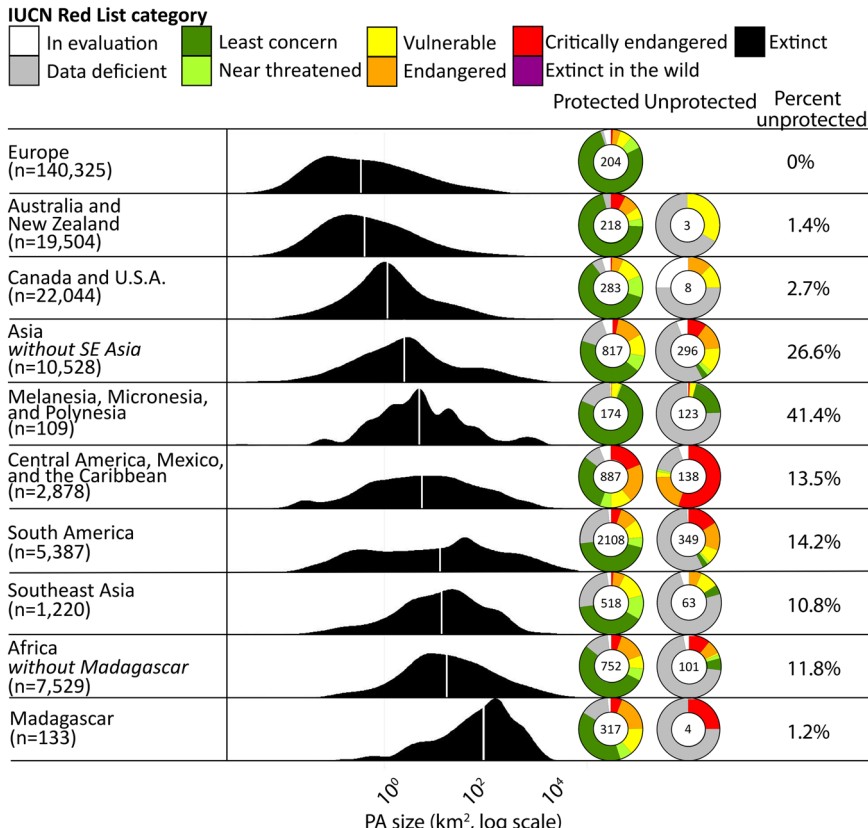

**Fig. 5 | Protected area size and threat status distribution by global regions.** From left to right, each global region shows a smoothed frequency histogram of PA sizes, a pie chart of the conservation status of protected amphibians, a pie chart for the conservation status of unprotected amphibians, and the percentage of unprotected species. The vertical white line on each smoothed density histogram represents its median value. The number of unprotected and protected species is shown on each pie chart. Regions are ordered by PA median size.

and Mexico are all within the top-ten countries in terms of their number of amphibian species, threatened species, and unprotected amphibian species.

## Discussion

Several studies over the last few decades have concluded that amphibians are underrepresented on the global PA network. In the present study, we integrate 778 new amphibian species range maps and 121,505 new PAs since the last similar assessment to find that the proportion of totally unprotected amphibian species has not improved over the last 19 years[4,6,12]. Important in understanding how a growing network has not resulted in improved coverage is the high rate of continued taxonomic discovery in Amphibia[19]. Though the number of global PAs increased from 24,993 in 1980 to 189,720 in 2020, the number of named amphibian species increased from 4318 to well over 8400 species over the same time period. However, we must also draw attention to the trend that the number of new amphibian-containing PAs– and amphibian-containing microreserves– has declined since the early 2000s, and the rate at which new amphibian species are added to the network has slowed since 2015 (Fig. 2a).

Our assessment of amphibian coverage by the current PA network forms different conclusions from the statistics reported earlier last year by Mi and colleagues[13], but differs substantively in focus and approach. First, we do not make assumptions about the status of historically-reported PAs that have subsequently been withheld from public release. Second, we do not impose a threshold percentage range coverage to consider an amphibian

included in the PA network, allowing us to compare our 'unprotected species' category with the 'gap species' reported by Rodrigues, Andelman, et al., 2004[4], and the 'unrepresented species' reported by Nori et al., 2015[12]. Given that the proportion of a species' habitat that must be preserved to promote its persistence varies widely based on factors like reproductive and dispersal strategy[40]--which exhibit particularly startling diversity in class Amphibia[41,42]--any threshold we might select would be arbitrary and a poor representation of the biological reality of a large part of amphibian diversity. Our selection means that we can provide a clean upper-bound estimate of the coverage that the World Database of Protected Areas (WDPA) provides for global amphibian diversity– where we can unambiguously state that the coverage provided by the WDPA leaves more species insufficiently protected than our estimate conveys– but by that same token does not imply that amphibians designated as "covered" are effectively protected. Third, we use a database of expert-curated amphibian range maps rather than ad hoc maps generated from accumulated occurrence points, resulting in the inclusion of an additional 1691 amphibian species ranges. Critically, our more complete dataset addresses the bias reported by Mi and colleagues towards the exclusion of small-range endemic species. Our more complete representation of small-range amphibians is particularly important, given our finding that a significantly smaller set of range sizes can, indeed, be considered a characteristic of class Amphibia (Fig. 1).

We find that unprotected species have a 6.2% higher chance of being threatened with extinction than protected species (Fig. 3). Encouragingly, land-based conservation efforts have in some senses become more targeted over time, with some PAs now being created in zones where they can benefit many threatened amphibian species (Fig. 2b). This trend is not yet pervasive enough that year of establishment is retained in our best-supported models of how proportion of threatened amphibians in a PA respond to PA characteristics (Table S2). Still, owing to these well-placed PAs, the number of threatened amphibian species left unprotected by our global PA network has decreased from 411 to 399 since 2004, while the proportion of all threatened amphibian species left unprotected has decreased from 26.6% to 18.4% (Fig. 3)[4]—a particularly important trend considering that amphibian populations were found to decline at threefold slower rates inside versus outside PAs in an analysis of 159 globally-distributed populations with time series available[43].

An important caveat is that our study finds that data deficient (DD) amphibians continue to be highly underrepresented by the PA network (Fig. 3). DD amphibians are significantly more likely to fall into threatened IUCN statuses (VU, EN, CR) than amphibians that have already been listed in non-DD categories by the IUCN[44,45]—with perhaps 85% of DD amphibians likely to be imperiled[46]. Therefore, our estimate that 35.8% of unprotected amphibians are currently threatened with extinction is a lower-end estimate of the actual value. For the purposes of conservation planning, it may be appropriate to assume DD amphibians are threatened until more information is gathered, though species that have already been designated as threatened can still be prioritized.

A major way that current land-based conservation efforts show taxonomic bias is in the assumption that PAs cannot be small if they are to be meaningful. Amphibians are implicitly neglected by this assumption, as are other endangered taxa being pushed to the brink by land use change[17,18]. We find that microreserves can host high amphibian species richness comparable to the largest global PAs (Fig. 4a), and that new microreserves increase amphibian representation in the PA network faster than new larger-sized PAs (Fig. 4b). This result is remarkable considering that our dataset was the existing database of registered PAs, so does not represent how efficiently microreserves could augment the coverage of amphibian diversity if regularly placed with small-range endemics in mind. We should note here that there is no standard definition of what constitutes a microreserve across the literature[47,48], so established this 10 km$^2$ threshold size for amphibian microreserves to particularly suit the distribution of possible amphibian range sizes (Fig. 1).

The conservation value of even very small PAs has already been recognized for plants[48]. Here, we argue that a greater recognition of the conservation value of microreserves may help reverse a worrying trend: the steep decline in the rate of new, amphibian-containing PA establishment within the PA network since 2000 (Fig. 2a). We find that many, spatially distributed PAs are best for improving the network's coverage of small-range endemic taxa with scattered ranges like amphibians (Fig. 4b; also see ref. 49): a growing PA network increases its coverage of amphibian diversity faster through the addition of smaller PAs, despite the fact that amphibian species richness in an individual PA tends to increase with its size (Table S3). To clarify, we do not advocate the downsizing of existing PAs– an increasingly common and problematic practice[50]—or that an increasing proportion of new PAs should be microreserves, given that they are already by far the most common size category of new PAs (Fig. 2a). We also do not envision microreserves as capable of promoting the indefinite persistence of the species they host, unless their habitat quality is maintained and they are part of an integrated approach that promotes stewardship of the surrounding matrix, supporting the ecological integrity of the patch and important species processes (e.g., dispersal, feeding, or overwintering)[32,33,51]. Instead, we conceive of microreserves as an important tool to more equitably represent different taxa within global PAs. Ideally, well-placed and well-managed microreserves will function as capillaries, promoting connectivity across 'landscapes that work for biodiversity and people'[52] and supporting the long-term functioning of the larger global PA network[33,53].

Microreserves must be placed strategically if they are to provide added value for amphibian conservation. We demonstrate that a microreserve of <10 km$^2$ could cover all or most of the distributional range of many amphibian species (e.g., microendemics, Fig. 1a), and that this is particularly true of threatened amphibians (Fig. 1B). Species with small ranges are frequently characterized by low local abundances[54], putting them at a higher risk of global extinction[55] and making their small ranges particularly important for inclusion in the PA network. In other cases, due to the extent of land conversion, tiny patches may be all that remains of once broader distributions[56,57]. Microreserves could be used to increase the PA network's coverage of point localities for data deficient or newly described amphibians when they are known from only a single point locality in cases where land conversion pressure is high.

Beyond microendemic amphibians, microreserves might also play an important role in protecting important source populations for amphibian species that exist in metapopulations, in protecting populations identified as being bastions of genetic diversity within a wider range[58], in protecting critical and endangered habitat types like wetlands used in breeding[59,60], or increasing the climatic niche representation of PAs within species' range to promote the preservation of evolutionary processes[61]. Microreserves could also be deployed to protect strings of habitat patches along climate migration corridors[62]. Admittedly, using microreserves in these ways implies a transformation of current, accepted concepts in PAs design. Beyond the plant conservation literature, microreserves currently appear in the literature almost exclusively for PA creation in urban-adjacent zones[47,48,63], often for recreation, whereas we propose to strategically deploy microreserves directly for biodiversity conservation.

In certain countries (Supplementary Fig. 1) and larger lobal regions (Fig. 5), the addition of microreserves would yield a particularly important conservation benefit. The areas of the world richest in small-range amphibian endemics, data deficient amphibians, and newly described amphibians (e.g. Southeast Asia, South America) correspond to regions where the median size of existing protected areas is, on average, much larger (Fig. 5). Regions of the world characterized by the greatest disparity between the proportion of threatened amphibian species existing within versus entirely outside of their PAs also tend to have larger median PA sizes (Central America, Mexico, and the Caribbean; and South America; Fig. 5). Thus, complementing the existing PA network in these regions with targeted microreserves to capture threatened amphibian species could be particularly transformative to their amphibian conservation landscape.

Mexico provides a compelling example of a country with rich opportunities to transform the biodiversity coverage of their PA network through the addition of targeted microreserves[64], particularly as land use change has

already been recognized as the most common threat to Mexican amphibians[65]. In our analysis, Mexico was in the global top ten countries in terms of species richness, endangered species richness, and total number of species currently having no overlap with the existing PA network– a status it shared with Colombia, China, Peru, and Ecuador (Supplementary Fig. 1). We found that about a quarter (25.5%) of Mexican amphibian species are left entirely unprotected by the current PA network– a result very similar to that obtained in a previous, country-specific analysis (23.7%)[66], despite our inclusion of an additional 46 amphibian species and 407 protected areas. Mexico has many microendemic amphibian species that are intrinsically well-suited to be protected by microreserves (Supplementary Fig. 2), as is frequently true for regions at lower latitudes.

Our estimates of where to establish microreserves for the greatest biodiversity gains is limited by current weaknesses of the World Database of Protected Areas (WDPA), though it is the largest and most complete database aggregating information on global protected areas. Private PAs, which tend to be smaller and in some respects of disproportionate biological importance relative to government-managed PAs, are under-reported in the WDPA[67], with only 20% of records in the database currently listed as non-governmental. Amphibian-rich Peru, which has reported more privately protected PAs within the WDPA than any other country (28,795 km²)[68], emerges in our analyses as being a country with some of the highest amphibian species diversity in its microreserves (Fig. 4a; PAs indicated). Better reporting of private PAs in the WDPA would facilitate better global gap analyses for the conservation of amphibians and other taxa with small range size.

To improve biodiversity conservation of species with small ranges, our results can be integrated into several important initiatives that provide information needed to support strategic microreserve design. For instance, the Alliance for Zero Extinction maintains a database of discrete sites serving as the last refuge of Endangered or Critically Endangered species[69]. The evolutionarily distinct globally endangered (EDGE) framework allows conservation planners to integrate considerations of phylogenetic distinctness[70], and a spatial prioritization approach that additionally incorporates endemism and anthropogenic pressures on a site has also been proposed[71]. In the U.S.A., the Priority Amphibian and Reptile Conservation Areas project[72] is conducting regional assessments to identify critical sites for herpetofaunal conservation based on species rarity, species richness, and landscape integrity.

Most stories about amphibian conservation reference the ongoing sixth mass extinction of global biodiversity and highlight the need for urgent conservation action. However, our study focuses attention on an encouraging note for protected area prioritization. As humanity unites in ambitious land-based conservation goals for the near future, it is a pivotal moment to revisit our assumptions about how very small PAs are valued and placed. Assuming by default that only larger PAs can conserve biodiversity will result in worse conservation outcomes for many taxa with restricted distributional patterns, not just amphibians. Based on our analyses, we propose that the placement of new microreserves is considered as carefully as the placement of their larger counterparts. This action could add significant amphibian conservation value to the PA network. Establishing targeted, biodiversity-motivated microreserves across the world could help protect thousands of threatened and endemic species, source populations that can shore up larger metapopulations, point localities of data deficient and newly described species, small but critical habitats, and strings of habitat along climate migration corridors.

## Methods
### Data acquisition
We used amphibian range maps from AmphibiaWeb and the International Union for the Conservation of Nature, or IUCN (available for 7094 species– over 83% of named amphibian species)[19,26,37]. For mammals and reptiles, we used ranges from the IUCN and the Global Assessment of Reptile Distributions group (10,811 reptiles and 5,850 mammals)[26,38,73,74]. For birds, we used ranges for 10,487 species from BirdLife International that

excluded species they consider sensitive[39,75] and joined all range polygons for each species, as they were originally separated into 'resident', 'breeding season', 'non-breeding season', 'passage', and 'seasonal occurrence uncertain' components. We acquired species' conservation status from the IUCN Red List of Threatened Species[26]. For amphibians, we included expert-curated provisional statuses[37].

We used the 240,999 PA polygons in terrestrial biomes from the World Database of Protected Areas (WDPA) database[3,76], trimming away any portions that overlapped marine habitats. The Russian Federation, Estonia, Saint Helena, Ascension, Tristan da Cunha, and China withhold all or part of their PA spatial data from public release[3], and we do not make assumptions about the current status of PAs previously reported to the WDPA and later withdrawn. Polygons of PAs that overlapped with each other were merged. We removed two polygons by searching for records that included the text "not protected", "degazetted," "proposed," "recommended," "in preparation," or "unset". We do not filter out WDPA based on their designated IUCN-protected area management category—referent to the objectives of PAs and what kind of activities can take place in them—in our analyses, except in our generalized linear models (as described below). It should be noted that for all analyses in which area of PAs is used, we use PA terrestrial area as reported by the WDPA (PA area less its marine area, i.e. GIS_AREA - GIS_M_AREA). These WDPA areas are calculated using an equal-area projection, to avoid area distortion near the poles.

### Vertebrate terrestrial range sizes
We estimated species distribution sizes from GIS polygon vectors. We compared ranges between all taxonomic groups first with a Kruskal-Wallis rank sum test, as data did not meet assumptions for an ANOVA, followed by tests between each taxon pair using two-sided Wilcoxon rank sum tests with continuity correction (Table S1). We performed the same tests for threatened or extinct members of these taxa only (including IUCN Red List categories VU, EN, CR, EW, and EX; Table S1). We visualized differences between all species of each taxa with a smoothed density histogram (Fig. 1a) and between threatened species using box and whisker plots (Fig. 1b).

### Overlap of PAs and amphibian ranges
To determine which amphibian species overlapped with a PA, we used QGIS 3.20 and reprojected the PA and amphibian range shapefiles in EPSG: 3857. This equal-angle projection was selected because preserving shapes of amphibian ranges and protected areas at a local scale is important for accurate overlap analysis. We took the intersection to generate lists of amphibians overlapping with each PA, with no minimum area threshold enforced. We calculated overlap statistics for both species that are threatened and not threatened, generating lists of species that are protected and unprotected (Fig. 3).

We visualized the history of counts of amphibian-containing PAs established since 1860, both overall and for microreserves only, with barplots. To this figure, we added two lines: the first represented the cumulative amphibian species coverage of the WDPA over time, and the second represented the cumulative amphibian species coverage of only microreserves in the WDPA over time (Fig. 2a). To understand how PA age might impact its conservation value, we used a hexbin heatmap to illustrate the relationship between the year of establishment of each PA and the proportion amphibian species it contained that were threatened (Fig. 2b). We also built a series of binomial family generalized linear models (GLMs) to explore how the proportion of amphibian species a PA contains that are threatened responds to PA characteristics and their interactions: degrees of latitude of the PA centroid from the equator, logarithmic PA area, and the WDPA data columns of IUCN protected area management category and establishment year. In order to include IUCN-protected area management category in our analysis, we removed PAs designated as 'Not Applicable', 'Not Assigned', or 'Not Reported' in our analysis. We used GLMs since the data violated the assumptions of classic linear regression, employing a binomial family GLM given that the response variable was a decimal value between 0 and 1. Model selection was performed by comparison of the

Akaike Information Criterion[77] between the full suite of models considered, including a null model, and McFadden's pseudo-r2 was calculated for these models (Table S2).

We used a second hexbin heatmap to describe how the size of amphibian-containing PAs relates to its total amphibian species richness (Fig. 4a). We used negative binomial GLMs to explore how the total amphibian species richness in a PA responded to the following PA characteristics and their interactions: degrees of latitude of the PA centroid from the equator, logarithmic PA area, IUCN protected area management category (once again excluding PAs designated as 'Not Applicable', 'Not Assigned', or 'Not Reported'), and PA establishment year. We employed GLMs as once again the data violated the assumptions of classic linear regression, and a negative binomial GLM given that the response variable was overdispersed count data. Model selection was performed by comparison of the Akaike Information Criterion[77] between the full suite of models considered, including a null model, and McFadden's pseudo-r2 was calculated for these models (Table S3).

To understand the impact of PA size on accumulated amphibian diversity, we categorized PAs into size classes: 0–10 km² ($n = 208,496$ PAs; 221,453 km² total area covered), 10–100 km² ($n = 27,762$; 948,694 km²), 100–1,000 km² ($n = 11,975$; 4,184,932 km²), 1,000–10,000 km² ($n = 3045$; 10,249,221 km²), and 10,000–100,000 km² ($n = 430$; 13,509,431 km²). We resampled PAs from a given size class with replacement until the cumulative area sampled reached the size of the total WDPA database in these size categories (29,113,730 km²). As each new PA was added, the cumulative number of unique amphibian species represented in the growing set was recorded. For each PA size class, this protocol was repeated 1000 times, and the mean number of cumulative species at each successive sampling stage was calculated. These mean values were used to create growth curves for each PA size class, with the x-axis scaled to represent equal area added and the y-axis representing the total number of amphibian species. We plotted the growth portion of these curves to compare the marginal benefit of adding PAs of different sizes to network coverage of amphibian diversity (Fig. 4b).

### Overlap of PA polygons, amphibian ranges, and geographic regions

To understand regional differences in PA size, and how well amphibian richness and threatened amphibian richness are represented in the WDPA network, we used the following biogeographic regions significant to the amphibian richness and endemism: Africa (excluding Madagascar); Asia (excluding SE Asia); Australia and New Zealand; Canada and the U.S.A.; Central America, Mexico, and the Caribbean; Europe; Madagascar; Melanesia, Micronesia, and Polynesia; and Southeast Asia (Brunei, East Timor, Indonesia, Malaysia, Philippines). For each region, we plotted a smoothed frequency histogram of PA size and graphed the total species in that region with respect to its conservation status of protected and unprotected amphibian species (Fig. 5).

To highlight countries of high conservation interest, we generated lists that ranked the top countries based on total, threatened, and unprotected amphibian species richness. Amphibian alpha-richness and threatened species richness were calculated with the range polygons used in this analysis and converted to a raster based on counts of overlapping polygons (implemented in R, raster v3.4). We selected Mexico as a case study to highlight how PA network and amphibian diversity interact at a country level (Supplementary Fig. 2).

### Statistics and reproducibility

Statistical analyses of comparative taxon distribution sizes, generalized linear model construction and selection, and all other data manipulations and visualizations were conducted in R using workflows documented on our GitHub (see Code availability statement). Our sampling was comprehensive of all data in the referenced publicly available databases, with the small, necessary exclusions documented in the methods above. The intermediate datatypes we derived from these public datasets are documented on Dryad[78] (see Data availability statement) to support reproducibility.

### Reporting summary

Further information on research design is available in the Nature Portfolio Reporting Summary linked to this article.

### Data availability

Data used in our analysis have been uploaded to DataDryad[78] (doi:10.5061/dryad.1c59zw429).

### Code availability

All data analysis and visualization was performed in Quantum GIS v3.2, ESRI ArcGIS v10.8, and in R v4.1.1 using libraries stringr v1.4.0, dplyr v1.0.7, plyr v1.8.6, tidyr v1.1.3, lessR v4.1.4, forcats v0.5.1, data.table v1.14.2, hexbin v1.28.2, ggridges v0.5.3, ggplot2 v3.3.5, raster v3.4., scales v1.1.1, nortest v1.0-4, MASS v7.3-54, pscl v1.5.9, car v3.0-11, and cowplot v1.1.1. Scripts to generate our analysis are available at https://github.com/AmphibiaWeb/amphibian-pa.

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

## Acknowledgements
We are indebted to the AmphibiaWeb GIS team in the Biodiversity Informatics Lab at the Museum of Vertebrate Zoology, with students supported by the Undergraduate Research Apprentice Program at the University of California, Berkeley. We thank Kevin Dang, Noelani Fixler, Alexandra Perkins, Yuerou Tang, Ziyue Wang, and Zoe Yoo for range-mapping from 2020–2021, and Yun Deng for computational advice. This manuscript was greatly improved through the thoughtful comments of several diligent reviewers.

## Author contributions
E.S., J.C., J.O., V.T.V., A.C., and M.K. designed the study. J.C. and J.O. processed the data. E.S. performed the analysis. E.S., J.C., J.O., and M.K. drafted the manuscript. All authors discussed the results, contributed critically to the drafts, and gave final approval for publication.

## Competing interests
The authors declare no competing interests.
