## [Peer review file · Communications Biology]

Reviewers' comments:

Reviewer #1 (Remarks to the Author):

It's an interesting and valuable paper, it shows the importance of microreserves, I like reading it. While I'm afraid some issue should be resolve first. My majority of comments are as follows:

1. Because the author calculated species' and PAs' range, so I think the author should use an equal-area projection rather than equal-angle projection (e.g., EPSG:3857 used in this study) would be more accurate to calculate areas.

2. The paper measure species whether are protected or not using species is in PAs or not in. It can tell readers how many species are included in PA now, while it can't indicate whether species are effectively protected. Small PAs have a larger number than large PAs (e.g., 0-10 km² vs. 10,000-100,000 km²: 208,496 vs. 430), the whole small PAs together include more diverse habitats, and related diverse species, but it does not mean small PAs are more efficiency. For example, we can use 1km² PAs for each amphibian species to protect 7,094 amphibian species, so 7,094 km² protected areas is enough? I suggest the author could also calculate the proportion of species habitat range as another metric, and see how species are protected in different PAs.

3. Besides, as I know, the median area of PA is less than 1 km², the author can divide 0-10 km² into two classes, 0-1 km² and 1-10 km², it's just a suggestion. In addition, the PAs from WDPA have six classes, so the author could clarify which classes they used.

4. There is an update reptile distribution range datasets which includes 10,064 species, see <https://datadryad.org/stash/dataset/doi:10.5061/dryad.83s7k>

Small comments:

Line 54, delete the repeat number of reference9,10.

Reviewer #2 (Remarks to the Author):

I reviewed with great interest the manuscript "Small is big: Microreserves contain hidden value for amphibians". It is a well written paper, clear and although not new it is interesting. I enjoyed the reading, but I do have some concerns.

My major concern is the lack of a single analysis that supports the main statement that is "Microreserves contain hidden value for amphibians", and although I do agree with the previous statement I expected an analysis showing how would improve the protection with micro reserves, because the figure 4b only shows the species protected in microreserves. And this analysis is not complicated, choose random conserved fragments of 0–10 sq km and explore how will increase protection, and actually explore the "value" of this fragments.

I am specially surprised about the lack of literature review, particularly about Mexico when the

authors clearly wanted to single it out. The authors should read and cite the below references, but just to be clear these are only a few examples.

Frías-Alvarez, P., Zúñiga-Vega, J. J., & Flores-Villela, O. (2010). A general assessment of the conservation status and decline trends of Mexican amphibians. *Biodiversity and Conservation*, 19, 3699-3742.

Ochoa-Ochoa, L., Urbina-Cardona, J. N., Vázquez, L. B., Flores-Villela, O., & Bezaury-Creel, J. (2009). The effects of governmental protected areas and social initiatives for land protection on the conservation of Mexican amphibians. *PLoS One*, 4(9), e6878.

Juárez-Ramírez, M. C., Aguilar-Lopez, J. L., & Pineda, E. (2016). Protected natural areas and the conservation of amphibians in a highly transformed mountainous region in Mexico. *Herpetological Conservation and Biology*, 11(1), 19-28.

Suazo-Ortuño, I., Ramírez-Bautista, A., & Alvarado-Díaz, J. (2023). Amphibians and Reptiles of Mexico: Diversity and Conservation. In *Mexican Fauna in the Anthropocene* (pp. 105-127). Cham: Springer International Publishing.

Quintero-Vallejo, D. E., & Ochoa-Ochoa, L. M. (2022). Priorización y distribución de los anfibios en las áreas naturales protegidas de México. *Revista Mexicana de Biodiversidad*, 93, e933939-e933939.

Particular comments

L29 vulnerable terrestrial vertebrates

L54 delete double citation

L56 the authors are making a GENERAL statement and the reference is NOT adequate.

L63 reference 19 is almost 20 years old

L63 amphibians have not been on earth for 400my, those were the first terrestrial vertebrates. Amphibians have been on earth ~200my but less as Lissamphibia, which are the modern amphibians that actually live on earth today

L66 extinction rates...

L56 Again, the authors are making a GENERAL statement and the reference is NOT adequate, although I really like that paper in particular.

L91 it should be stated clearly where the data came from

L98-100 THIS IS OBVIOUS range size is the first aspect that is evaluated in order to establish the categories

L127-128 there is no analysis nor any reference to support this statement

L174-175 given the data used the authors should establish a threshold of coverage to determine presence, and it should be in reference to the range size of the species; exactly as percentage range coverage

L243-247 Please read the references mentioned

Reviewer #3 (Remarks to the Author):

Please see attached file

Response to Reviewers

Reviewer #1 (Remarks to the Author):

It's an interesting and valuable paper, it shows the importance of microreserves, I like reading it. While I'm afraid some issue should be resolve first. My majority of comments are as follows:

Thank you for reviewing our paper, and for your constructive comments to improve it. Responses to your suggestions are provided, one-by-one, below.

1. Because the author calculated species' and PAs' range, so I think the author should use an equal-area projection rather than equal-angle projection (e.g., EPSG:3857 used in this study) would be more accurate to calculate areas.

Thank you for this observation– we have made more explicit how the strengths of equal-area versus equal-angle projections were employed. To class protected area sizes, we used terrestrial protected area size as reported by the World Database of Protected Areas (PA area less its marine area, i.e. GIS_AREA - GIS_M_AREA), which is calculated using an equal-area projection. We use equal-angle projections for our overlap analyses, because it is critically important that shape of amphibian range and protected area polygons is locally accurate, such that overlaps are correctly analyzed. I have added further explanation of these points to the text so that readers can understand our process and decision-making as well as possible!

Line 414: “It should be noted that for all analyses in which area of PAs is used, we use PA terrestrial area as reported by the WDPA (PA area less its marine area, i.e. GIS_AREA - GIS_M_AREA). These WDPA areas are calculated using an equal-area projection, to avoid area distorsion near the poles.”

Line 432: “To determine which amphibian species overlapped with a PA, we used QGIS 3.20 and reprojected the PA and amphibian range shapefiles in EPSG: 3857. This equal-angle projection was selected because preserving shapes of amphibian ranges and protected areas at a local scale is important for accurate overlap analysis.”

2. The paper measure species whether are protected or not using species is in PAs or not in. It can tell readers how many species are included in PA now, while it can't indicate whether species are effectively protected. Small PAs have a larger number than large PAs (e.g., 0-10 km² vs. 10,000-100,000 km²: 208,496 vs. 430), the whole small PAs together include more diverse habitats, and related diverse species, but it does not mean small PAs are more efficiency. For example, we can use 1km² PAs for each amphibian species to protect 7,094 amphibian species, so 7,094 km² protected areas is enough?

Our study illustrates that establishment of microreserves (small PAs) could be an overlooked conservation tool that could augment ongoing conservation efforts, not replace large reserves. Because microreserves are small, they can be more easily located in diverse habitats (e.g., spread out geographically, elevationally, and across more biomes/ecosystems). We use amphibians as a threatened group that could greatly benefit from microreserves.

We certainly do not intend to advocate for an increasing proportion of new PAs to be microreserves. Already, a very large proportion of new PAs are microreserves (Fig. 2A). We hope

our reworking of the manuscript does a better job at highlighting this important point. For example, we have added additional detail to our existing disambiguation of our perspective in the Discussion, as follows:

Line 223: “Second, we do not impose a threshold percentage range coverage to consider an amphibian included in the PA network, allowing us to compare our ‘unprotected species’ category with the ‘gap species’ reported by Rodrigues et al., 2004, and the ‘unrepresented species’ reported by Nori et al., 2015. Given that the proportion of a species’ habitat that must be preserved to promote its persistence varies widely based on factors like reproductive and dispersal strategy (Fahrig, 2001)--which exhibit particularly startling diversity in class Amphibia (Crump et al., 2015; Smith and Green, 2005)--any threshold we might select would be arbitrary and a poor representation of the biological reality of a large part of amphibian diversity. Our selection means that we can provide a clean upper-bound estimate of the coverage that the WDPA provides global amphibian diversity-- where we can unambiguously state that the coverage provided by the WDPA leaves more species insufficiently protected than our estimate conveys-- but by that same token does not imply that amphibians designated as “covered” are effectively protected.”

Line 280: “However, we certainly do not advocate the downsizing of existing PAs-- an increasingly common and problematic practice (Watson et al., 2014). Neither do we advocate that an increasing proportion of new PAs should be microreserves, given that they are already by far the most common size category of new PAs (Fig. 2A). Finally, it is important to clarify that we do not envision microreserves as capable of promoting the indefinite persistence of the species they host, unless their habitat quality is maintained and they are part of an integrated approach that promotes stewardship of the surrounding matrix, supporting the ecological integrity of the patch and important species processes (e.g., dispersal, feeding, or overwintering)(Cushman, 2006; Hartel, Scheele, Rozyłowicz, Horcea-milcu, & Cogălniceanu, 2020; Volenec & Dobson, 2019). Instead, we conceive of microreserves as an important tool to more equitably represent different taxa within global PAs. Ideally, well-placed and well-managed microreserves will function as capillaries, promoting connectivity across ‘landscapes that work for biodiversity and people’ (Kremen and Merenlender, 2018) and supporting the long-term functioning of the larger global PA network(Catenazzi, 2015; Volenec & Dobson, 2019).”

I suggest the author could also calculate the proportion of species habitat range as another metric, and see how species are protected in different PAs.

This would definitely be one approach to a coverage analysis, and it is an approach that we considered when we designed our study. Not imposing a proportion-of-range or else a set area-based threshold was an explicit choice we made. I have included additional description of our reasoning behind this choice in the discussion:

Line 224: “Second, we do not impose a threshold percentage range coverage to consider an amphibian included in the PA network, allowing us to compare our ‘unprotected species’ category with the ‘gap species’ reported by Rodrigues et al., 2004, and the ‘unrepresented species’ reported by Nori et al., 2015. Given that the proportion of a species’ habitat that must be preserved to promote its persistence varies widely based on factors like reproductive and dispersal strategy (Fahrig, 2001)--which exhibit particularly startling diversity in class Amphibia (Crump et al., 2015; Smith and Green, 2005)--any threshold we might select would

be arbitrary and a poor representation of the biological reality of a large part of amphibian diversity. Our selection means that we can provide a clean upper-bound estimate of the coverage that the WDPa provides global amphibian diversity— where we can unambiguously acknowledge that the coverage provided by the WDPa leaves more species insufficiently protected than our estimate conveys.”

3. Besides, as I know, the median area of PA is less than 1 km², the author can divide 0-10 km² into two classes, 0-1 km² and 1-10 km², it's just a suggestion. In addition, the PAs from WDPa have six classes, so the author could clarify which classes they used.

It is a good point that we should explicitly discuss the IUCN category of PAs in our methods, so have done so in our revision.

e.g., from methods, line 445: “We also built a series of binomial family generalized linear models (GLMs) to explore how the proportion of amphibian species a PA contains that are threatened responds to PA characteristics and their interactions: latitude of the PA centroid, logarithmic PA area, PA IUCN category, and PA establishment year. In order to include IUCN category in our analysis, we removed PAs designated as ‘Not Applicable’, ‘Not Assigned’, or ‘Not Reported’ in our analysis.”

and corresponding example from results, line 127: “Over time, we also see that more PAs are being established in zones of high amphibian vulnerability (Fig. 2b). The best-supported model of how the proportion of threatened amphibian species in a PA respond to PA characteristics (latitude, ln(area), year of establishment, and IUCN category) retained latitude, logarithmic area, and the interaction between these terms (Table S2), with all coefficients being significant ($p < 0.001$).”

4. There is an update reptile distribution range datasets which includes 10,064 species, see <https://datadryad.org/stash/dataset/doi:10.5061/dryad.83s7k>

We now include the additional 2,418 species from GARD in our figure and analyses comparing taxon ranges.

Line 397: “For mammals and reptiles, we used ranges from the IUCN and the Global Assessment of Reptile Distributions group (10,811 reptiles and 5,850 mammals)(IUCN, 2021; Meiri et al., 2017).”

Small comments:

Line 54, delete the repeat number of reference9,10.

Done.

Reviewer #2 (Remarks to the Author):

I reviewed with great interest the manuscript “Small is big: Microreserves contain hidden value for amphibians”. It is a well written paper, clear and although not new it is interesting. I enjoyed the reading, but I do have some concerns. My major concern is the lack of a single analysis that supports the main statement that is “Microreserves contain hidden value for amphibians”, and although I do agree with the previous statement I expected an analysis showing how would improve the protection with micro reserves, because the figure 4b only shows the species protected in microreserves. And this analysis is not complicated, choose random conserved fragments of 0–10 sq km and explore how will increase protection, and actually explore the “value” of this fragments.

Figure 4B shows that microreserves performed better than larger reserves. This figure shows species protected in microreserves (0-10km²), as well as the amphibians protected in PAs of four additional size categories (10-100km², 100-1.000km², 1.000-10.000km², and 10.000-100.000km²). Sampling random plots of this size from the terrestrial surface of the globe would not better prove our point for two reasons. The first is that we do not advocate for the random placement of microreserves, but rather we urge that microreserves are consistently placed with as much careful thought for biodiversity as is invested in larger PAs. The second reason is that our figure, as currently produced, already represents the existing global set of microreserves– and so overwhelmingly already represents the non-directed placement of PAs with regards to amphibian diversity. We have done our best to make sure that these ideas are as clear as possible in the re-worked discussion.

e.g., Line 201: the intro to the discussion ends, “Critically, we show that the PA network’s coverage of amphibian diversity can be more rapidly augmented through the addition of

microreserves. Together, we use our findings to argue that effective amphibian conservation will require that we not discount the conservation potential of new microreserves. Instead, we urge that microreserves be deployed very strategically, particularly to capture small-range species that will otherwise get left-behind in land-based conservation efforts.”

Line 282: “We find that many, spatially distributed PAs are best for improving the network's coverage of small-range endemic taxa with scattered ranges like amphibians (Fig. 4b; also see Armsworth et al., 2018).”

Line 376: “Based on our analyses, we propose that the placement of new microreserves is considered as carefully as the placement of their larger counterparts.”

We explore the value of these fragments in a few ways beyond Fig. 4B. In Fig. 2B, we show that even very old microreserves are capable of hosting high amphibian species richness with a low proportion of threatened species. In Fig. 4A, we show that the species richness hosted by microreserves can nearly rival that of the largest global PAs. We have added to Fig. 2A a depiction of how the cumulative number of amphibian species covered by the PA network has grown over time if only microreserves are considered. We have also added formal analyses of the data depicted in Figures 2B and 4A, with full results shown in Table S2 and Table S3.

I am specially surprised about the lack of literature review, particularly about Mexico when the authors clearly wanted to single it out. The authors should read and cite the below references, but just to be clear these are only a few examples. / L243-247 Please read the references mentioned

Frías-Alvarez, P., Zúñiga-Vega, J. J., & Flores-Villela, O. (2010). A general assessment of the conservation status and decline trends of Mexican amphibians. *Biodiversity and Conservation*, 19, 3699-3742.

Ochoa-Ochoa, L., Urbina-Cardona, J. N., Vázquez, L. B., Flores-Villela, O., & Bezaury-Creel, J. (2009). The effects of governmental protected areas and social initiatives for land protection on the conservation of Mexican amphibians. *PLoS One*, 4(9), e6878.

Juárez-Ramírez, M. C., Aguilar-Lopez, J. L., & Pineda, E. (2016). Protected natural areas and the

conservation of amphibians in a highly transformed mountainous region in Mexico. *Herpetological Conservation and Biology*, 11(1), 19-28.

Suazo-Ortuño, I., Ramírez-Bautista, A., & Alvarado-Díaz, J. (2023). Amphibians and Reptiles of Mexico: Diversity and Conservation. In *Mexican Fauna in the Anthropocene* (pp. 105-127). Cham: Springer International Publishing.

Quintero-Vallejo, D. E., & Ochoa-Ochoa, L. M. (2022). Priorización y distribución de los anfibios en las áreas naturales protegidas de México. *Revista Mexicana de Biodiversidad*, 93, e933939-e933939.

Thank you for the suggested references. We have integrated additional findings from previous studies of Mexican amphibian diversity into our text. More generally, we have updated our manuscript with findings from a number of new publications that have emerged since our initial manuscript submission, providing additional context to our results.

Particular comments

L29 vulnerable terrestrial vertebrates

We have changed to 'the most vulnerable vertebrate class' (now line 29).

L54 delete double citation

Done.

L56 the authors are making a GENERAL statement and the reference is NOT adequate.

I think here you are referencing this sentence: “For example, the existing PA network serves amphibians particularly poorly¹³, such that they are the most underrepresented class of terrestrial vertebrates^{6,14}.” I have brought these three references together at the end of the sentence to make the strong support for this claim very clear, and have also added a fourth reference:

Line 60: “For example, the existing PA network serves amphibians particularly poorly, such that they are the most underrepresented class of terrestrial vertebrates(Mi et al., 2023; Rodrigues, Akçakaya, et al., 2004; Urbina-Cardona & Flores-Villela, 2010; Venter et al., 2014).”

L63 reference 19 is almost 20 years old

I have replaced this reference.

L63 amphibians have not been on earth for 400my, those were the first terrestrial vertebrates. Amphibians have been on earth ~200my but less as Lissamphibia, which are the modern amphibians that actually live on earth today

We add a reference to our statement that, "amphibians have existed on earth for nearly 400 million years" (now Line 70). This reference dates the origin of Amphibia to late in the Devonian period (~359-419 million years ago).

L66 extinction rates...

To make the language clearer and simpler have reworded from

"The rate of current declines set amphibians on track for extinction rates that exceed those estimated for previous mass extinctions"

to

"Current declines set amphibians on track for extinction rates exceeding those estimated for previous mass extinctions" (now Line 73).

L91 it should be stated clearly where the data came from

We now include the organizations responsible for managing this data and the respective citations in our results reporting (Line 104).

L98-100 THIS IS OBVIOUS range size is the first aspect that is evaluated in order to establish the categories

It is true that this finding may not appear at first surprising ("Finally, within class Amphibia, threatened species had a smaller median range size than non-threatened species (Wilcoxon rank sum test; $p < 0.001$)."). However, we feel that this is not obvious, as it is not universally true. It is indeed common for amphibian assessments to use Red List criteria related to size, but the Red List assess species trends only over the past 10 years. Therefore, if a small-range species is recovering it could still be moved to 'near threatened', even if they biologically underwent important population bottlenecks in the past.

L127-128 there is no analysis nor any reference to support this statement

The data to support this statement is all visualized in Figure 4A. I reconfigure in-text mentions of the elements in this figure to guide the reader better, and also have revised the figure caption to help readers better interpret the figure.

Line 157: "Although most microreserves (<10 km²) are currently located in areas of low amphibian richness (Fig. 4A), we find that microreserves are also able to capture areas of high richness (annotations i, ii, & iii)"

L174-175 given the data used the authors should establish a threshold of coverage to determine presence, and it should be in reference to the range size of the species; exactly as percentage range coverage

This would definitely be one approach to a coverage analysis, and it is an approach that we considered when we designed our study. Not imposing a proportion-of-range or else a set area-

based threshold was an explicit choice we made. I have included additional description of our reasoning behind this choice in the discussion:

Line 225: “Second, we do not impose a threshold percentage range coverage to consider an amphibian included in the PA network, allowing us to compare our ‘unprotected species’ category with the ‘gap species’ reported by Rodrigues et al., 2004, and the ‘unrepresented species’ reported by Nori et al., 2015. Given that the proportion of a species’ habitat that must be preserved to promote its persistence varies widely based on factors like reproductive and dispersal strategy (Fahrig, 2001)--which exhibit particularly startling diversity in class Amphibia (Crump et al., 2015; Smith and Green, 2005)--any threshold we might select would be arbitrary and a poor representation of the biological reality of a large part of amphibian diversity. Our selection means that we can provide a clean upper-bound estimate of the coverage that the WDPa provides global amphibian diversity– where we can unambiguously acknowledge that the coverage provided by the WDPa leaves more species insufficiently protected than our estimate conveys.”

Reviewer#3

General comments

The title of this ms is certainly very attractive and invite to a promising lecture. The role of protected areas (isolated or as part of a network) to achieve species conservation is a interesting subject. Analyses that help to improve the current approach (eg. increasing efficacy in creating new locations, resources allocation, etc) are very welcome especially in a time of global (including climate) change.

Thank you for reviewing our manuscript. We look forward to addressing each of your concerns, line-by-line, below.

However, as it develops, it is becoming clear, in my opinion, that the approach lacks originality and supporting entity. For instance, the subject of microreserves for conserving biodiversity it is not original at all (it has been pioneered in plants since early 2000s , ref #36). Also, the analysis of amphibian conservation based on protected areas was already carried out by Nori et al (2015, ref #15, among others eg. SánchezFernandez and Abellán 2015), although Steigerwald et al. declare some differences in their approach and conclusions (Lines 173-177).

While microreserves have indeed been touted as important tools in plant (and insect) conservation, we believe that those focusing on the conservation of vertebrates continue to discount the conservation potential of very small protected areas. Our work illustrates that microreserves could be highly beneficial to the most threatened class of vertebrates, amphibians. We think that amphibians make a perfect case study, though indeed there are many microendemic mammals, reptiles, and even bird species– particularly on islands and mountaintops– that would also benefit from our greater appreciation of microreserves.

At the end, it would rest to know a) which is, in authors opinion, the “hidden value” of the microreserves for amphibians, and b) the supporting information of their objectives, in particular toward the implementation of microreserves as a land-based complementary effort for amphibian conservation. In sum, the ms is interesting but need some attention in several sections. The discussion section seems speculative in some parts.

Our central thesis is that the conservation value of very small protected areas is discounted, but that if we fail to use them to protect the ranges of very small range endemics we will fail to integrate those species into the global PA network. We have clarified this message throughout the manuscript, including re-wording and re-organizing many segments, particularly in the Discussion.

e.g., we have changed our title from “Small is big: Microreserves contain hidden value for amphibians” to “Small is big: Microreserves are an important tool for amphibian conservation”.

Line 38: Abstract concludes, “We propose that stemming global biodiversity loss requires that we seriously consider the conservation potential of microreserves, using them to capture small-range endemics that may otherwise be omitted from the PA network entirely.”

Line 264: Section title is now, “Small-range endemics lose when microreserves are not placed strategically”

Line 379: Conclusion ends, “Based on our analyses, we propose that the placement of new microreserves is considered as carefully as the placement of their larger counterparts. This action could add significant amphibian conservation value to the PA network. Establishing targeted, biodiversity-motivated microreserves across the world could help protect thousands of threatened and endemic species, source populations that can shore up larger metapopulations, point localities of data deficient and newly described species, small but critical habitats, and strings of habitat along climate migration corridors.”

To address your comment about speculative language in the discussion, I have reviewed the discussion for any occurrence of speculative language, making the tone certain when appropriate and ensuring any claims a reader might doubt are backed with sufficient reference material:

e.g., “Here, we *suggest* that a greater recognition of their conservation value may help reverse a worrying trend” becomes Line 279: “Here, we *argue* that a greater recognition of the conservation value of microreserves may help reverse a worrying trend”.

“Therefore, our estimate that 35.8% of unprotected amphibians are currently threatened with extinction is an underestimation and *should be considered* the lower-bound estimate of the actual value.” becomes Line 258: “Therefore, our estimate that 35.8% of unprotected amphibians are currently threatened with extinction *is* a lower-end estimate of the actual value.”

“We highlight countries (Fig. S1) and larger global regions (Fig. 5) where the addition of microreserves *could* yield the greatest conservation benefit.” becomes Line 325: “We highlight countries (Fig. S1) and larger global regions (Fig. 5) where the addition of microreserves *would* yield the greatest conservation benefit.”

Addition of subsection titles to Discussion, to make our salient points pithy and clear: Line 206 “Despite some wins for threatened species, global PAs continue to under-represent amphibian

diversity”; Line 263 “Small-range endemics lose when microreserves are not placed strategically”; Line 320 “Amphibians in the Neotropics have the most to gain from targeted microreserve creation”.

I agree with authors in questioning the status quo of current PAs as the main land-based conservation strategy, and in many other aspects (e.g. economic and social constraints affecting PA placement, strong taxonomic bias in species representation in current PAs, need to expand the PA designation). At the same time, I would like to encourage authors to consider the need to implement a new paradigm for habitat and species conservation beyond traditional PAs. Certainly, microreserves (some 20ha or so, both private or governmental) may be a transitional way to expand the view.

I have provided more detail to the segment on how to support a PA network where microreserves are regularly deployed for biodiversity conservation as follows:

Line 289: “Finally, it is important to clarify that we do not envision microreserves as capable of promoting the indefinite persistence of the species they host, unless their habitat quality is maintained and they are part of an integrated approach that promotes stewardship of the surrounding matrix, supporting the ecological integrity of the patch and important species processes (e.g., dispersal, feeding, or overwintering; Cushman, 2006; Hartel et al., 2020; Volenec & Dobson, 2019). Instead, we conceive of microreserves as an important tool to more equitably represent different taxa within global PAs. Ideally, these PAs will function as capillaries, promoting connectivity across ‘landscapes that work for biodiversity and people’ (Kremen and Merenlender, 2018) and supporting the long-term functioning of the larger global PA network(Catenazzi, 2015; Volenec & Dobson, 2019).”

We focus on the scope we have delineated for ourselves—why should microreserves be taken seriously as a tool for amphibian conservation— so do not provide additional commentary on other conservation strategies unrelated to PAs.

Comments on specific sections and issues

Title:

I suggest to include the word “conservation” in the title. Ej. Small is big: Microreserves contain hidden value for amphibian conservation

We have renamed: “Small is big: Microreserves are an important tool for amphibian conservation”.

Keywords: please clarify 30x30 (eg. “30x30” Target)

Done. We have replaced “30x30” with "30x30" Target.

Introduction:

One thing that seems quite obvious is to provide a clear definition of what is, for authors, a microreserve (e.g. “ A plant micro-reserve (PMR) is a small plot of land -up to 20ha – there is no minimum size that is of peak value in terms of plant richness, endemism or rarity. A PMR is a permanent, statutory reserve given over to long-term monitoring of endangered wild plant species and vegetation types” Kadis et al.

2013: Plant micro-reserves: From theory to practice. Utopia Publishing. 194 pp). Soon in the Abstract (line 30) they write: microreserves (< 10 km²), and later (line 127) they mentioned again (1-10 km²). In my opinion, this amount of territory only can be considered “micro” under a relative perspective, and it is far distant of the 20ha (i.e. 0,2 km²), the maximum considered for plants (cfr. Kadis et al 2013).

Thanks so much for pointing out this typo (the 1-10km² was a mistake). We explicitly define 'microreserve', both in our revised abstract and introduction.

Line 29: “Focusing on amphibians, the most vulnerable vertebrate class, we illustrate the conservation value of microreserves, a term we employ here to refer to reserves of <10 km².”

Line 89: “Therefore, we explore the idea that strategically-placed reserves of 10km² or less, here termed as 'microreserves', could drastically enhance the value of the PA network for amphibians.”

We use this term out of convenience for referring to the smallest size category in our size classification system. We do recognize the need to be explicit in our definition, given that there is no standardized definition of the size defining a ‘microreserve’. Previous authors have also used the term with various other upper thresholds in mind (e.g. Ikauniece et al. 2011; Vandergast et al. 2009).

Line 235: “We should note here that there is no standard definition of what constitutes a microreserve across the literature (Vandergast et al., 2009, Kadis et al. 2013), so established this 10 km² threshold size for amphibian microreserves to particularly suit the distribution of possible amphibian range sizes (Fig. 1).”

Vandergast, A.G., Lewallen, E.A., Deas, J. et al. Loss of genetic connectivity and diversity in urban microreserves in a southern California endemic Jerusalem cricket (Orthoptera: Stenopelmatidae: Stenopelmatus n. sp. “santa monica”). *J Insect Conserv* 13, 329–345 (2009). <https://doi.org/10.1007/s10841-008-9176-z>

Ikauniece, S. (2011). Protection of forest habitats outside Natura 2000—experience and problems in Latvia. *Legal Aspects of European Forest Sustainable Development*, 60.

Materials and Methods:

Results:

A large part of the reported results (sections #1 and #2: “On average amphibians have smaller ranges than other terrestrial vertebrates”, “The rate at which amphibians containing PAs are created is declining...”) are, in my opinion, rather collateral to paper’s objectives. The third and fourth sections (“PA networks cover amphibian diversity more rapidly through the addition of smaller PAs...”, “Amphibians in the Neotropics...”) are on target.

The results of sections 1 and 2 are both important to the perspective we are presenting in this piece, but we are grateful for the opportunity to make the centrality of these findings clearer. We have worked additionally on the framing of these findings in our abstract and discussion, as well as the setup for these analyses in our introduction, and hope that this makes our motivation for exploring these topics clearer to our readers.

e.g., from the Abstract, “We demonstrate that microreserves could protect a substantial portion of many amphibian ranges, particularly threatened species, by proving something

previously assumed: amphibians generally have smaller ranges than other terrestrial vertebrates.” becomes Line 33: “By proving something previously assumed—that amphibians generally have smaller ranges than other terrestrial vertebrates—we demonstrate that microreserves could protect a substantial portion of many amphibian ranges, particularly threatened species.”

At the end of the Introduction, we add Line 90, “We formally test whether amphibians do, in fact, have smaller range sizes on average than other taxa, as is often assumed.”.

We added an introductory paragraph to the Discussion, reviewing all findings and how they fit together, including Lin 195: “Here, we show that amphibians continue to be underrepresented by the global PA network, yet fewer amphibian-containing PAs are being created over time, such that the rate at which amphibian diversity is being integrated into the global PA network has recently lagged. We confirm that amphibians generally have smaller ranges than other terrestrial vertebrate classes, as previously assumed.”

Line 129: please check correspondence with Figure 4 (there is not Figure 4A, nor 4c). Something similar appears with Figure 2B (line 130), and again Figure 4B (line 133) or Figure 4A (line 199). Please revise all these (text and figures).

We have changed this to i, ii, and iii in both the text and the figures. We have also added explicit reference to the roman numerals (i, ii, & iii) to our existing explanation of the annotations on the figure caption.

Interesting results are, in fact, depicted in Figure 4. Nevertheless, the analysis needs to be completed. For instance, it seems appropriate to use the data of species richness in Figure 4a comparing the so-called “microreserves” by authors (<10 km²) vs. the rest of PAs (larger than 10 km²).

I have conducted more formal analyses to accompany both Figures 2B and 4A, the detailed results of which are now communicated in Tables S2 and S3 and summarized in the text. While it is a well-established ecological relationship that larger habitat fragments can support more species, what we are show in Figure 4A and our discussion of it is that microreserves are capable of supporting high amphibian species richness. We hope that this perspective is clearer in the revised manuscript!

Several additional reported results are weakening, in my opinion, the idea of promoting the creation per se of new microreserves. First, the number of PAs by size reported by authors in Figure 4b: considering a total of 251.708 PAs, microreserves (<10 km²) are largely the most common, reaching 82,8% (208.496). That is, as noted by authors, even considering the limitations of current WDPA database.

We certainly do not intend to advocate for an increasing proportion of new PAs to be microreserves. Already, a very large proportion of new PAs are microreserves (Fig. 2A). Rather, we argue that microreserve placement has historically been not considered important in the way that the placement of larger reserves has been, such that microreserves are not manifesting their conservation potential. We hope our reworking of the manuscript does a better job at highlighting these important point. For example:

Line 280: “However, we certainly do not advocate the downsizing of existing PAs– an increasingly common and problematic practice (Watson et al., 2014). Neither do we advocate that an increasing proportion of new PAs should be microreserves, given that they are already by far the most common size category of new PAs (Fig. 2A). Finally, it is important to clarify that we do not envision microreserves as capable of promoting the indefinite persistence of the species they host, unless their habitat quality is maintained and they are part of an integrated approach that promotes stewardship of the surrounding matrix, supporting the ecological integrity of the patch and important species processes (e.g., dispersal, feeding, or

overwintering)(Cushman, 2006; Hartel, Scheele, Rozyłowicz, Horcea-milcu, & Cogălniceanu, 2020; Volenec & Dobson, 2019). Instead, we conceive of microreserves as an important tool to more equitably represent different taxa within global PAs. Ideally, well-placed and well-managed microreserves will function as capillaries, promoting connectivity across ‘landscapes that work for biodiversity and people’ (Kremen and Merenlender, 2018) and supporting the long-term functioning of the larger global PA network(Catenazzi, 2015; Volenec & Dobson, 2019).”

Line 317: “Beyond the plant conservation literature, microreserves currently appear in the literature almost exclusively for PA creation in urban-adjacent zones(Delaney, Busteed, Fisher, & Riley, 2021; Laguna, Ballester, & Deltoro, 2013; Vandergast et al., 2009), often for recreation, whereas we propose to strategically deploy microreserves directly for biodiversity conservation.”

Second, also reported by authors (line 125) that there is a positive but weak relationship between current PA size and amphibians species richness.

We have shared the results of our statistical analyses to accompany both Figures 2B and 4A, the detailed results of which are now communicated in Tables S2 and S3 and summarized in the text (Line 130 and Line 152). While it is a well-established ecological relationship that larger habitat fragments can support more species, what we show with Figure 4A and our discussion of it is that microreserves are capable of supporting high amphibian species richness. We hope that this perspective is clearer in the revised manuscript!

Third, the underestimation of the role of private PAs (lines 250-257), also in the WDPA, that makes me to think that the real situation may better than is being reported here. I suggest new, updated data, from other countries (e.g. Costa Rica). / Lines 249-252: refers to previous comments.

I agree that it could theoretically be more robust if our work included a more comprehensive accounting of private protected areas. Unfortunately, additional databases including shapefiles for private PAs are not available. For example, I can find that there were 213 private protected areas in Costa Rica as of 2014 (Stolton et al. 2014), and what must be then an incomplete list of their names (Red Costarricense de Reservas Naturales, 2023), but no associated shapefiles. If private PA-specific databases exist and are simply difficult to locate, it would be extraordinarily time-intensive to combine data from multiple sources while avoiding duplication, as databases will have no key in common so must be manually checked in the case of each possible duplicate. Other analyses of global protected areas that we have identified also have used the WDPA, addressing limitations to their study based on its shortcomings in the same way as we have done. We feel use of such an accessible, transparent database additionally increases the reproducibility of our work.

Red Costarricense de Reservas Naturales. (2023). Lista de reservas. Website: www.reservasnaturales.org. Accessed: 15 Dec 2022.

Stolton, S., Redford, K. H., Dudley, N., & Bill, W. (2014). The futures of privately protected areas. IUCN, Gland, Switzerland.

Discussion:

Line 183: authors mentioned that global PA network has decreased from 411 to 399 (12, i.e. 0,029 %). In my opinion, this figure indicate that it remains rather stable since 2004.

It is the count of unprotected amphibian species left unprotected that has decreased from 411 to 399, not the number of global PAs. I reworted the text here to enhance clarity (Line 249).

Line 214: In my opinion, this is the right and key question that should be considered here. From all comments above, seems that improving the current conservation approach using PAs is more a question of well-placed PAs than creating more of the type proposed microreserves (perhaps with the exceptions indicated in Figure 5).

We agree: we should be using microreserves to advantage in biodiversity conservation, and not placing them thoughtlessly due to the misapprehension that they are not useful conservation tools. We have refined language throughout the article to feel certain this point is not left ambiguous. e.g.

Line 38: Abstract concludes, “We propose that stemming global biodiversity loss requires that we seriously consider the conservation potential of microreserves, using them to capture small-range endemics that may otherwise be omitted from the PA network entirely.”

Line 264: Section title is now, “Small-range endemics lose when microreserves are not placed strategically”

Line 376: Conclusion ends, “Based on our analyses, we propose that the placement of new microreserves is considered as carefully as the placement of their larger counterparts. This action could add significant amphibian conservation value to the PA network. Establishing targeted, biodiversity-motivated microreserves across the world could help protect thousands of threatened and endemic species, source populations that can shore up larger metapopulations, point localities of data deficient and newly described species, small but critical habitats, and strings of habitat along climate migration corridors.”

In this figure (Fig. 5) I suggest:

a) Explain what are the numbers (n=140,325, n=19,504, and so on, I assume that is PA size, but it would be better to say so in the figure legend, and avoid using the letter “n” for these items),

The title now reads, Region

(total number of PAs)

With each subsequent region formatted,

Europe

(140,325)

b) Last column on the right: indicate “PA percent unprotected”, and

c) Include a new column on the right with absolute number of species

The column titles are now “Number and IUCN category of amphibians that are protected | unprotected”, to refer to the pie charts; and “Percent of species left unprotected”, to refer to the column of percentages at the far right.

IUCN Red List category

REVIEWERS' COMMENTS:

Reviewer #2 (Remarks to the Author):

I've read the author's responses of the manuscript "mall is big: Microreserves are an important tool for amphibian conservation" and I am satisfied, although I do not agree with certain choices. The authors have addressed all my concerns, including the statistical analyses requested. I approve this submission.

Reviewer #3 (Remarks to the Author):

The authors have convincingly addressed the comments and suggestions.